# Probing fractional statistics in quantum simulators of spin liquid Hamiltonians

**Shiyu Zhou[1], Maria Zelenayova[2,3,4], Oliver Hart[2,5],**
**Claudio Chamon[1] and Claudio Castelnovo[2]**

**1** Department of Physics, Boston University, Boston, MA, 02215, USA
**2** TCM Group, Cavendish Laboratory, University of Cambridge, Cambridge CB3 0HE, UK
**3** Max Planck Institute for the Science of Light, Staudtstraße 2, 91058 Erlangen, Germany
**4** Friedrich-Alexander-Universität Erlangen-Nürnberg,
Martensstraße 9, 91058 Erlangen, Germany
**5** Department of Physics and Center for Theory of Quantum Matter,
University of Colorado, Boulder, Colorado 80309 USA

## Abstract

Recent advances in programmable quantum devices brought to the fore the intriguing possibility of using them to realise and investigate topological quantum spin liquid phases. This new and exciting direction brings about important research questions on how to probe and determine the presence of such exotic, highly entangled phases in a noisy quantum environment. One of the most promising tools is investigating the behaviour of the topological excitations, and in particular their fractional statistics. In this work we put forward a generic route to achieve this, and we illustrate it in the specific case of $\mathbb{Z}_2$ topological spin liquids implemented with the aid of combinatorial gauge symmetry. We design a convenient architecture to study signatures of fractional statistics via quasiparticle interferometry, and we assess its robustness to diagonal and off-diagonal disorder, as well as to dephasing – effects that are generally pervasive in current quantum programmable devices. Interestingly, when turned on its head, our scheme provides a remarkably clear test of the 'quantumness' of these devices, with robust signatures that crucially hinge on quantum coherence and quantum interference effects, and cannot be mimicked by classical stochastic processes.



# 1 Introduction

A distinctive feature of topologically ordered quantum fluids [1] is the fact that they host excitations with fractional exchange statistics [2, 3]: when two identical quasiparticles trade positions, the many-body wavefunction can acquire a phase other than 0 (as for bosons) or $\pi$ (as for fermions). Quantum Hall liquids are examples of topologically ordered charged fluids with anyonic excitations [4, 5]. Experimental validation of fractional statistics in the quantum Hall effect was only recently accomplished [6, 7]; in particular, nontrivial exchange statistics were observed in Ref. [7] using an interferometric approach [8]. Excitations with fractional statistics are also predicted in quantum spin liquids – systems with topological but not conventional magnetic order [9, 10]. While efforts to unambiguously identify gapped spin liquids in naturally occurring materials are ongoing [11], the individual manipulation of quasiparticles needed to detect fractional statistics – as achieved in electronic quantum Hall systems – is not, in any evident way, within reach of being attained in magnetic materials, although recent theoretical efforts have advanced proposals of possible probes [12, 13].

Here instead, we present ways of realising and experimenting with quantum spin liquid states using time-independent Hamiltonians that are programmable in quantum devices, specifically focusing on devising ways to probe the exchange statistics of the quasiparticles. As we shall demonstrate, an interesting byproduct of this research direction is the identification of novel avenues to critically test quantum coherence in such programmable quantum devices. Moreover, since these devices – as much perhaps as real materials – necessarily operate at finite temperature and in presence of noise and disorder, in our work we will set the stage and provide at least initial answers to important questions such as: How much coherence (i.e., on what length and time scales) is needed to observe effects of anyonic statistics in noisy topological quantum environments? And how do signatures of fractionalisation fare in presence of disorder, noise and dissipation?

To carry out this scheme, one must first design topological spin liquid Hamiltonians with realistic (1- and 2-body) interactions that could be programmed in existing quantum devices. (For instance, programmable $ZZ$ couplings and uniform transverse fields are available in D-Wave annealers [14].) A key element of our work is the notion of combinatorial gauge symmetry (CGS), which allows us to attain such quantum Hamiltonians while preserving *exact* gauge symmetries, present for example in solvable quantum spin liquid models, like Kitaev's $\mathbb{Z}_2$ toric code.

There are two types of excitations in a $\mathbb{Z}_2$ quantum spin liquid: spinons and visons; and their excitation gaps differ substantially in the case of Hamiltonians programmed as in Ref. [15]. The reason is that the vison gap is only perturbatively generated in powers of the ratio of the transverse field $\Gamma$ over the $ZZ$ interaction $J$ (and $\Gamma < J$ is strictly required to realise a $\mathbb{Z}_2$ spin liquid phase). The spinon gap, on the other hand, is of order $J$ and on its own it lands the system on a classical $\mathbb{Z}_2$ spin liquid state in which one can directly observe the loop structure of the 8-vertex model [15]. This wide separation of scales between the spinon and vison gaps is common whenever one relies on a perturbative expansion to obtain the local gauge generators in an effective theory.

Refs. [16, 17] identified signatures of the mutual statistics of the spinons and visons that could be observed in the regime where temperature is larger than the vison gap but smaller than the spinon gap. In this regime the visons stochastically appear on random plaquettes because their energy of formation is much smaller than temperature. In the presence of quantum kinetic terms (such as a transverse field not weak compared to thermal noise), the spinons acquire dynamics at a scale much faster than that of the visons, and they effectively quantum diffuse in a background of randomly placed visons. Because of the mutual sermionic statistics between the two types of quasiparticles, the quasi-static visons serve as sources of random $\pi$ fluxes, which lead to quantum interference corrections to the diffusion of the spinons in 2D. In principle these corrections could be detectable, but an actual implementation of these ideas in a programmable quantum device must incorporate other complications that may make comparison of the theory to the data difficult, such as sources of decoherence and disorder in the couplings, and embedding complexities.

Here we present a scheme to detect the mutual statistics of spinons and visons interferometrically. This proposal takes into account the aforementioned practical complications. The main idea is not to work on a 2D geometry, but in a quasi-1D one, in which case the effect of the $\pi$ fluxes is magnified, leading to a sharper signature of the mutual statistics.

Because many programmable quantum devices are 'black boxes' to users, it is imperative that we develop benchmarks and tests with the purpose of building confidence that what is programmed reflects the physical system we intend to probe. One such test is to study the effect of disorder and dephasing on the observables of these experiments, and to benchmark quantum against classical behaviour. Such analysis allows us to have a comparative reference to better understand also the degree of disorder and decoherence in the device.

It is important to note that, if successful, these experiments would constitute a direct observation of the statistics of visons and spinons in a static Hamiltonian setting. They would therefore be complementary to those carried out in Ref. [18], using a 219-atom programmable quantum simulator using Rydberg atoms, and in Ref. [19], through the evolution via a quantum circuit running on Google's lattice of superconducting qubits. In these systems the topological state is prepared dynamically, either by quasi-adiabatic state preparation [18] or by application of quantum gates [19]. Both are truly remarkable experiments that achieve the dynamical preparation of a quantum spin liquid state. The goal of our work is to instead design these same phases as the *ground state* of *static* Hamiltonians. Along this path, theoretical principles such as combinatorial gauge symmetry are developed, and deployed.

In Sec. 2, we present a quasi-1D ladder model with the combinatorial gauge symmetry and its effective mapping to a quasi-1D $\mathbb{Z}_2$ ladder by explicitly showing their local $\mathbb{Z}_2$ gauge symmetry generators. We then discuss how the mutual sermionic statistics between spinons and visons – two types of quasiparticle excitations of both models leads to the fragmentation of spinon dynamics. In Sec. 3, we numerically simulate the spinon dynamics in the absence and presence of visons by mapping it to a tight-binding model in a 1D chain where a single spinon hops between sites and the quasi-static visons locate on the bonds. We further show the robustness of spinon dynamics against dephasing and disorders – effects that are prevailing in

current noisy intermediate-scale quantum (NISQ) devices, and compare it with classical particle diffusion with disorders. We then include a study on how mobile visons affect the spinon dynamics. In Sec. 4, we detail the mapping between the CGS ladder and the $\mathbb{Z}_2$ ladder. We study their low energy spectrum and find a one-to-one correspondence between the couplings from the two models, and examine how the mutual semionic statistics in the $\mathbb{Z}_2$ ladder arise in the CGS model. In Sec. 5, we elaborate on how probing the mutual statistic of a quasi-1D $\mathbb{Z}_2$ ladder can be used to test the quantumness. Finally, we conclude in Sec. 6.

In Appendix A we further offer an example of how our scheme can be used to embed a Hamiltonian with an exact $\mathbb{Z}_2$ gauge symmetry on a commercial D-wave device [15]. Our results suggest that the operating temperatures and time scales in DW-2000Q are just outside the range needed to observe coherent quasiparticle propagation in the $\mathbb{Z}_2$ quantum spin liquid phase. Future developments of this architecture, as well as possibly current versions of other architectures may however be suitable to observe the phenomena discussed in this work.

## 2 Probing mutual statistics of visons and spinons on a CGS ladder

The system we focus on is the 2-leg ladder in Fig. 1(a), with a gauge spin $\sigma$ on each link $\ell$, and four matter spins $\mu$ on each site $s$. The matter spins are coupled to their four neighbouring gauge spins according to the Hamiltonian

$$H = -\sum_s \left[ J \sum_{\substack{a \in s \\ \ell \in s}} W_{a\ell}\, \mu_a^z \sigma_\ell^z + \Gamma_0 \sum_{a \in s} \mu_a^x \right] - \Gamma_0 \sum_\ell \sigma_\ell^x, \tag{1a}$$

where $a \in s$ labels the four matter spins on site $s$, $\ell \in s$ labels the four bonds connected to site $s$, and $W$ is the $4 \times 4$ Hadamard matrix

$$W = \begin{pmatrix} -1 & +1 & +1 & +1 \\ +1 & -1 & +1 & +1 \\ +1 & +1 & -1 & +1 \\ +1 & +1 & +1 & -1 \end{pmatrix}. \tag{1b}$$

We shall henceforth refer to this system as as a 'CGS ladder' (where CGS stands for combinatorial gauge symmetry [20]).

This Hamiltonian has a local $\mathbb{Z}_2$ symmetry, generated by an operator $G_p$ at plaquette $p$, see Fig. 1(a), that both flips the two gauge spins around the plaquette, and flips and permutes the states of the matter spins on the two stars neighbouring the plaquette:

$$G_p = F_p^L\, B_p\, F_p^R. \tag{2}$$

Here $B_p$ is the product of the $x$-components of the two gauge spins on the top and bottom links of the plaquette $p$. The operators $F_p^L$ and $F_p^R$ flip and permute matter spins on the two stars to the left and right ($L$ and $R$, respectively) of the plaquette. Specifically, $F_p^L$ permutes matter spins $\mu_1 \leftrightarrow \mu_2$ and $\mu_3 \leftrightarrow \mu_4$, and flips $\mu_1$ and $\mu_2$; $F_p^R$ permutes matter spins $\mu_1 \leftrightarrow \mu_2$ and $\mu_3 \leftrightarrow \mu_4$, and flips $\mu_3$ and $\mu_4$, as illustrated in Fig. 1(a). Explicitly, these operators can be written in terms of SWAP gates and spin-flip operators:

$$F_p^L = \tfrac{1}{2}(\mathbb{1} + \boldsymbol{\mu}_1 \cdot \boldsymbol{\mu}_2)\, \mu_1^x \mu_2^x\, \tfrac{1}{2}(\mathbb{1} + \boldsymbol{\mu}_3 \cdot \boldsymbol{\mu}_4),$$
$$F_p^R = \tfrac{1}{2}(\mathbb{1} + \boldsymbol{\mu}_1 \cdot \boldsymbol{\mu}_2)\, \mu_3^x \mu_4^x\, \tfrac{1}{2}(\mathbb{1} + \boldsymbol{\mu}_3 \cdot \boldsymbol{\mu}_4).$$

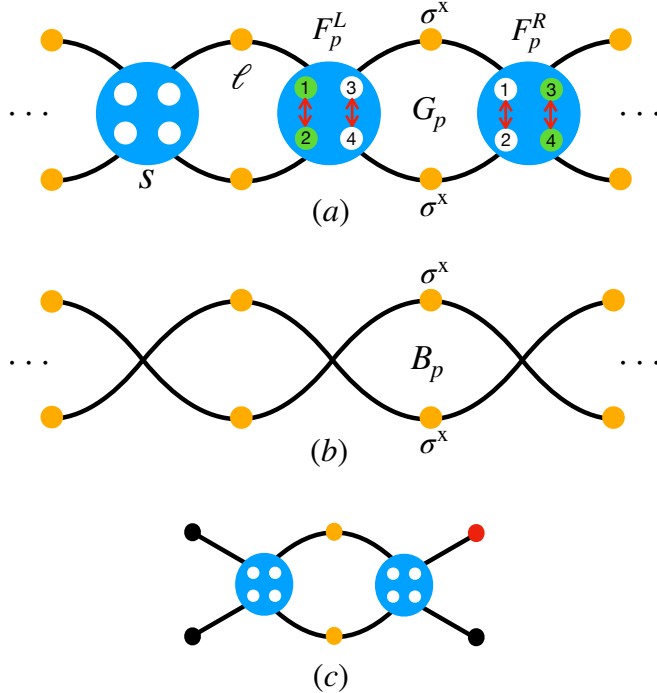

Figure 1: (a) Geometry of the CGS ladder. Each vertex $s$ has four on-site matter spins (coloured in white) and four gauge spins (coloured in orange) on the links $\ell$, and they interact according to Eq. (1). Neighbouring sites share two gauge spins to form a quasi-1D ladder. Local gauge symmetry operators $G_p = B_p \, F_p^L \, F_p^R$ of Eq. (2) act on plaquettes, labelled by $p$. The operators $F_p^L$ and $F_p^R$ permute (according to the doubled-headed arrows in red) and flip matter spins (coloured in green) on the two stars to the left and right [$L$ and $R$] of $G_p$. (b) Geometry of the $\mathbb{Z}_2$ ladder. It has the same symmetry as the CGS ladder in (a). The products of the $x$-component of the two spins on the top and bottom branches define the generators of gauge transformations, $B_p$. We refer to this model, with Hamiltonian Eq. (3), as a $\mathbb{Z}_2$ ladder. (c) Geometry of two coupled CGS stars used to extract, by exact diagonalisation, the effective $\mathbb{Z}_2$ ladder couplings from the CGS model, thus relating models (a) and (b). The effective spinon hopping amplitude can be obtained from the energy splittings in the spectrum of this small system with the boundary spins pinned to be in odd parity with $3+$ (black) and $1-$ (red) gauge spins (see Sec. 4).

One can verify that $B_p^2 = \left(F_p^L\right)^2 = \left(F_p^R\right)^2 = 1$, and thus $G_p^2 = 1$, and that $[G_p, G_{p'}] = 0$. Correspondingly, the combinatorial gauge symmetry that gives rise to the $\mathbb{Z}_2$ symmetry of the quantum spin liquid system proposed in Ref. [20] is also present in this quasi-1D model.

The operators $G_p$ have eigenvalues $\pm 1$, which serve as conserved quantities of $H$. A vison excitation is said to exist at plaquette $p$ when $G_p$ has eigenvalue $-1$. A spinon excitation is a direct violation of the low energy conditions that minimise the dominant $J$ term in a star $s$ in the Hamiltonian Eq. (1). Both excitations are gapped. (The vison gap appears only perturbatively, once the transverse field is turned on.)

The CGS ladder depicted in Fig. 1(a) can be effectively mapped onto a quasi-1D $\mathbb{Z}_2$ lattice gauge model (a $\mathbb{Z}_2$ ladder), depicted in Fig. 1(b). Only the gauge spins $\sigma$ on the links $\ell$ are retained as degrees of freedom, and the first term in the Hamiltonian can then be written in terms of the usual toric code star operators involving the four bonds connected at a given

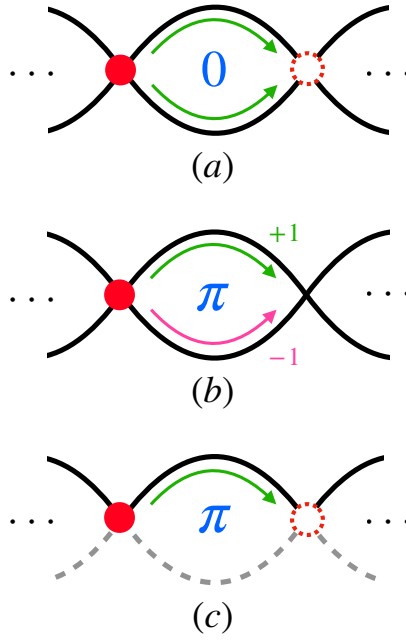

Figure 2: Schematics of spinon propagation along the quasi-1D $\mathbb{Z}_2$ ladder in the absence/presence of visons (labeled as a $0/\pi$ flux in blue at the centre of the plaquette shown in the figure). (a) A single spinon (labelled as a red dot) initialised on the left site, and a 0 flux state for the plaquette immediately on its right, allow it to hop constructively across the top and bottom paths of the ladder, shown by the green arrows, to the right neighbouring site, depicted as a red dashed circle. (b) A vison is now located on the central plaquette, equivalent to a $\pi$ flux threading it. Its presence gives rise to destructive interference (opposite phases) of the two paths (top shown in green with $+1$ phase and bottom shown in pink with $-1$ phase) due to the mutual statistics of spinons and visons, and consequently the spinon cannot hop to the site on the right. Namely, the presence of visons blocks the propagation of spinons, and cuts the ladder into disconnected segments. (c) The gauge spins on the bottom legs of the ladder are frozen (illustrated by grey dashed lines), e.g., pinned by an applied field that is large compared to the other scales in the problem. The pinning field on the bottom leg forces spinons to propagate only along the top leg of the ladder, effectively removing any interference and therefore any interplay between spinons and visons. In this case, spinons can hop freely along the chain.

site [21],

$$H = -\lambda \sum_s \prod_{\ell \in s} \sigma_\ell^z - \frac{\Gamma}{2} \sum_\ell \sigma_\ell^x, \tag{3}$$

where $\lambda$ and $\Gamma$ are appropriate energy scales [the choice of a factor of $1/2$ in front of $\Gamma$ is explained below Eq. (5)]. This $\mathbb{Z}_2$ ladder shares the same local gauge symmetry (here generated by $B_p$) and quasiparticle excitations – spinons (violations of the low energy conditions that minimise the dominant $\lambda$ term in a star $s$) and visons (corresponding to eigenvalues $B_p = -1$) – of the CGS ladder. The correspondence is investigated quantitatively in Sec. 4, where the couplings of the effective $\mathbb{Z}_2$ ladder, $\lambda$ and $\Gamma/2$, are obtained from the microscopic couplings $J$ and $\Gamma_0$ of Eq. (1) through a numerical comparison of the spinon gap and hopping amplitudes. In Sec. 4 we also discuss the emergence of semionic statistics between spinons and visons in the CGS ladder.

Consider the regime $T \lesssim \Gamma \ll \lambda$, where temperature is much smaller than the spinon gap ($\sim \lambda$) and generally smaller than the transverse field, but much larger than the vison gap ($\sim \Gamma^2/\lambda$); and consider an initial state with a single spinon localised on a given site. The spinon number is effectively conserved in this regime,[1] and therefore the spinon behaves as a tight-binding particle evolving in time due to the transverse field $\Gamma$, propagating through a background of thermally distributed visons. Because of the small matrix elements associated to vison dynamics, the visons are quasi-static on the time scale of spinon hopping. This is the same physical regime considered in Refs. [16,17] for 2D, but now in quasi-1D.

The virtue of the quasi-1D geometry proposed here is that the presence of visons halts the spinon propagation altogether. Indeed, there are two paths for the spinon to move between adjacent sites, as shown in Fig. 2(a): along the top or the bottom link of a plaquette (akin to the two arms of an interferometer). The presence of a vison on a plaquette gives rise to perfect destructive interference of the two paths, because of the $\pi$ phase shift due to mutual semionic statistics.[2] Therefore the presence of visons cuts the quasi-1D system into finite disconnected subsystems: spinons can only propagate along segments containing no visons. Since the density of visons is close to a half in the limit of large temperature compared to the vison gap, the mean free path of the spinons should be of the order of one lattice spacing.

This fragmentation of the spinon propagation, which trivially leads to a localised wave function, can be comparatively tested against the scenario where a large longitudinal field is applied on, say, the bottom links of the 2-leg ladder, see Fig. 2(b). The presence of this field favours the gauge spins on the bottom links to point in the Zeeman-preferred direction, thus suppressing spinon propagation along them. This in turn reduces the destructive interference and increases the spinon mobility along the upper links of the ladder. In the limiting case where the spins on the bottom links are pinned and the spinon can only propagate along the top path, there is no quantum interference at all and free tight-binding propagation is unimpeded. This delocalisation of the spinon as the bottom leg becomes polarised provides a distinctive signature of the semionic statistics between spinons and visons.

In any currently available NISQ devices, certain degrees of noise or disorder are unavoidable, due for instance to non-zero operating temperature or cross-talk between qubits [25]. Hence, in the next section, we study in detail the spinon propagation in the background of thermally distributed visons with and without effects of noise and disorder, and compare them with classical diffusion. At last, we examine how vison dynamics affects the spinon propagation.

# 3  Study of the $\mathbb{Z}_2$ ladder

In order to account for the effects of a noisy environment, we focus for simplicity on the limit where the CGS ladder is well approximated by the $\mathbb{Z}_2$ ladder model (i.e., a toric-code-like Hamiltonian), which in turn can be described as a model of tight-binding spinons (with hopping amplitude $\Gamma$) on the sites of a 1D chain, with quasi-static stochastic visons on the bonds. The vison energy scale is taken to be negligibly small, and their density is $\rho_v = 1/2$ throughout our work.

We focus mainly on the single spinon behaviour in a fixed vison background, therefore neglecting dissipative processes corresponding to quasiparticle creation and annihilation events. These are only accounted for, in an effective way in the form of classical stochastic vison creation and annihilation events, in Sec. 3.4.

---

[1] The long spinon lifetime in the limit of small $\Gamma/\lambda$ is analogous to the exponentially long lifetime of doublons in the Hubbard model in the large-$U$ limit [22–24].

[2] Notice that moving a spinon around a vison corresponds to exchanging the two quasiparticles twice. The semionic statistical angle upon exchanging them is $\pi/2$.

Dephasing in the spinon dynamics can be thought of having two effects: on the one hand, phase coherence is lost as a function of time as the spinon propagates along the ladder; on the other hand, dephasing introduces a small random phase mismatch as the spinon hops to an adjacent site along the top and bottom links of the ladder. For computational convenience, we separate these two effects and treat them as if they were independent of one another. We implement the first effect by including a dephasing Lindblad term, of strength $\gamma$, in our spinon tight-binding model. Since we are not interested in straightforward quantum oscillatory behaviour of the wavefunction (see the discussion in Sec. 5), we set for convenience $\gamma = \Gamma/2$. Different values of $\gamma$ could easily be studied but we do not expect them to qualitatively alter the conclusions of our work with respect to the long time behaviour.

As discussed in Sec. 2, visons disconnect an ideal 1D chain into separate segments. This distinctive effect due to mutual semionic statistics is a promising experimentally viable route to a signature of quantum spin liquid behaviour, as well as a test of quantum mechanical coherence in the system. The second effect of dephasing however introduces a phase mismatch that alters the otherwise perfect destructive/constructive interference in the presence/absence of visons. We mimic this effect in an approximate way by introducing static random off-diagonal disorder in the spinon tight-binding model. (Incidentally, the fact that signatures of the presence of visons survive up to off-diagonal disorder as strong as $\sim \Gamma/2$, as illustrated in Fig. 4, is a justification a posteriori of the choice of Lindblad dephasing $\gamma = \Gamma/2$.)

For completeness, we also include in this section a study of the effects of a static random onsite potential (diagonal disorder) for the spinons, as well as the effects of (stroboscopic) stochastic dynamics of the visons background. Finally, it will be important to contrast the behaviour of the quantum system to that of a classical particle performing diffusive random walk on a disordered 1D chain – if we are indeed to devise reliable signatures of quantum interference effects.

One final note is in order. Given that our tight-binding particles are in fact emergent quasiparticles in a spin system, one ought to account for noise coupling to the spins. This leads to what was dubbed 'dephasing with strings attached' [26]. In the parameter range of interest in our work however (namely, for $\gamma = \Gamma/2$ and for the size and time scales considered here) we find that the strings only lead to minimal quantitative differences that can be safely ignored.

### 3.1 Base model

Our effective clean reference model is a spinon hopping on a 1D chain, with amplitude $\Gamma = 1$ and Lindblad dephasing $\gamma = \Gamma/2$, governed by the dynamical equation

$$\dot{\rho}_{ss'} = -i [H, \rho]_{ss'} - \gamma (1 - \delta_{ss'}) \rho_{ss'}, \tag{4}$$

where $s, s'$ label the position of the spinon, and its Hamiltonian reads

$$H = -\Gamma \sum_{s=1}^{L-1} \left( \hat{b}_s^\dagger \hat{b}_{s+1} + \text{h.c} \right), \tag{5}$$

with $\Gamma$ being the hopping amplitude and $\hat{b}_s^\dagger, \hat{b}_s$ the bosonic operators creating and annihilating a spinon on site $s$. Notice that the hopping amplitude $\Gamma$ of a spinon hopping in a single chain is twice the strength of a spinon hopping on a ladder with two legs in Eq. (3). In the clean limit considered here, the presence of a vison on a bond switches the corresponding hopping term $\Gamma$ to zero.

We simulate a chain of length $L = 25$ with open boundary conditions, governed by Eq. (4). We implement a vectorized form of the density matrix [27] leading to ordinary differential

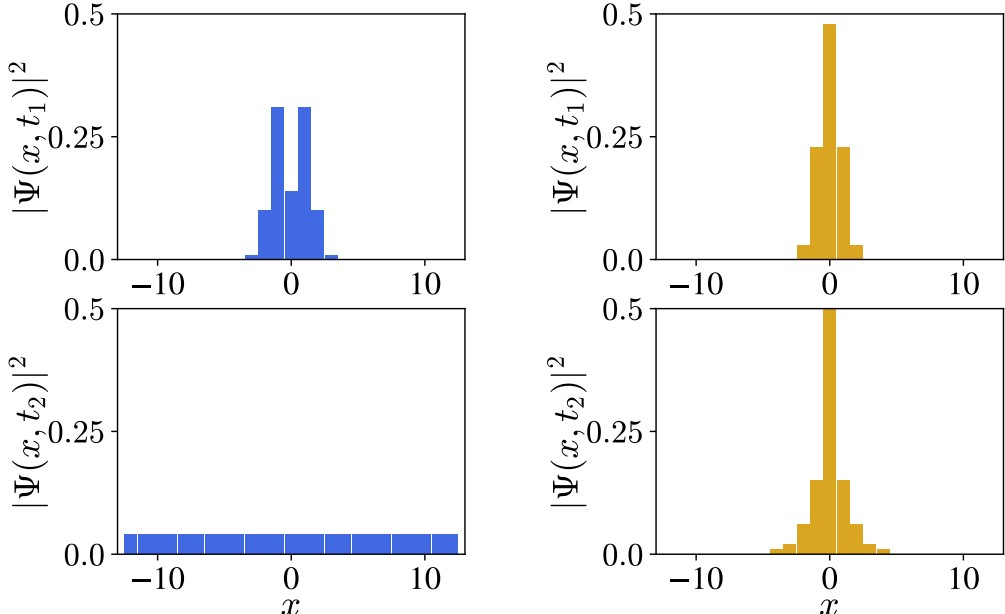

Figure 3: Histograms of the spinon wavefunction $|\Psi(x,t)|^2$ as a function of position $x$ (in units of lattice spacing) at different times, for our base model Eq. (4) on a chain of length $L = 25$, after the spinon is initially localised in the middle of the system. Top panels: time $t_1 = 1$ in units of $1/\Gamma$. Bottom panels: time $t_2 = 100$. The right vs left panels illustrate the remarkable effect of the presence vs absence of mutual semionic statistics, respectively. The wavefunction profile $|\Psi(x,t)|^2$ has been averaged over $10 \cdot 2^{10}$ infinite temperature vison configurations with average vison density $\rho_v = 1/2$.

equations, which we then solve numerically to obtain the spinon wavefunction $\Psi(x,t)$. When visons are present in the system (equivalently, in presence of mutual semionic statistics) we average our results over $10 \cdot 2^{10}$ infinite temperature vison configurations with average vison density $\rho_v = 1/2$. The resulting behaviour of $|\Psi(x,t)|^2$ is illustrated in Fig. 3.

The effect of mutual statistics between spinons and visons is dramatic. For example, a spinon initially localised on a given site along the chain can only relax to a wave function with finite support (Fig. 3), that can be computed exactly (see App. B). This is in stark contrast with the case where mutual statistics is switched off (equivalently, the visons are removed from the system), and the wave function can relax and become uniformly delocalised across the chain.

## 3.2 Disorder

We next investigate how robust this signature of mutual semionic statistics and quantum spin liquid behaviour is to the presence of both diagonal and off-diagonal disorder. Diagonal disorder is expected to lead to localisation of the spinon, possibly inducing a behaviour similar to disconnecting the chain into segments due to the presence of visons. (This same point has already been investigated in higher dimensions [17,28], where it was shown that the effect of disorder is significantly different, and weaker, than the effect of visons and mutual statistics.) Off-diagonal disorder also generically leads to localisation in 1D systems; moreover, it can allow hopping across bonds that would otherwise be disconnected because of perfect destructive interference in presence of visons.

We discuss here the effects of diagonal and off-diagonal disorder on the spinons, whilst keeping the visons randomly distributed and static. However, one should note that what we

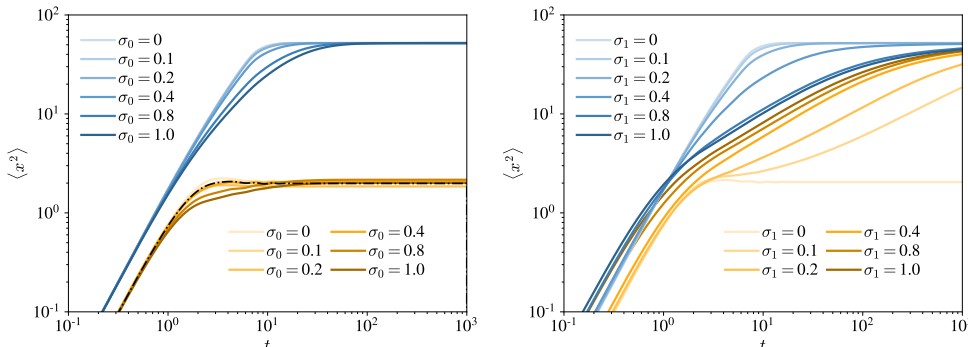

Figure 4: Comparison of the effects of random onsite disorder (zero mean and standard deviation $\sigma_0$), top panel, and random hopping amplitudes (zero mean and standard deviation $\sigma_1$), bottom panel. In all cases the behaviour of $\langle x^2 \rangle$ (in units of lattice spacing) vs time $t$ (in units of $1/\Gamma$) was computed with dephasing $\gamma = \Gamma/2$, hopping $\Gamma = 1$, and averaged over $2^{10}$ disorder realisations for system size $L = 25$; the blue-palette curves correspond to propagation in the absence of visons, and the orange-palette ones to the presence of visons (in which case, the random vison configuration was also changed with each disorder realisation). In the top panel we also show the semi-analytical solution of the case without disorder (dash-dotted black line), detailed in App. B.

refer to here as off-diagonal terms correspond to onsite energies for the visons, whereas diagonal terms correspond to vison hopping amplitudes, whose effects we do not model. The former are unlikely to affect our conclusions, as random pinning energies do not introduce vison correlations. The latter on the other hand introduce random quantum fluctuations in the vison configuration; we refer the reader to Sec. 3.4 for an approximate study of their effect.

Fig. 4 shows the results in presence of random disorder in the onsite energy (diagonal disorder, $w_s^{(\mathrm{d})}$, Gaussian distributed with zero mean and standard deviation $\sigma_0$) and random hopping amplitudes (off-diagonal disorder, $w_{s,s'}^{(\mathrm{o})}$, again Gaussian distributed with zero mean and standard deviation $\sigma_1$),

$$H_{\mathrm{dis}} = \sum_{s=1}^{L-1} w_{s,s'}^{(\mathrm{o})} \left( \hat{b}_s^\dagger \hat{b}_{s+1} + \mathrm{h.c} \right) + \sum_{s=1}^{L} w_s^{(\mathrm{d})} \hat{b}_s^\dagger \hat{b}_s . \tag{6}$$

We then proceed to solve Eq. (4) as before, but this time we add Eq. (6) to the Hamiltonian in Eq. (5). We compute the average squared displacement $\langle x^2 \rangle$ (where position is measured in units of lattice spacing) as a function of time (in units of $1/\Gamma$), after the spinon is initialised at the centre of the chain, contrasting the behaviour with and without visons (which is equivalent to switching on or off the mutual semionic statistics). The results are then averaged over $2^{10}$ disorder realizations in which visons are randomly distributed with density $\rho_v = 1/2$ on the bonds of the chain.

We consider the two types of disorders separately for convenience, and we checked that indeed their simultaneous presence does not lead to any significantly different cooperative behaviour. We include for comparison the semi-analytical solution of the case without disorder (see App. B).

The offset (notable at short times) between the case with and without visons is due to the destructive interference which takes place on a site immediately next to the initial position, i.e., the presence of a vison in a plaquette adjacent to the initial position cuts the chain and thus locks the spinon on the initial site. Correspondingly, we see that the offset vanishes for $w_{s,s'}^{(\mathrm{o})} \to 1$.

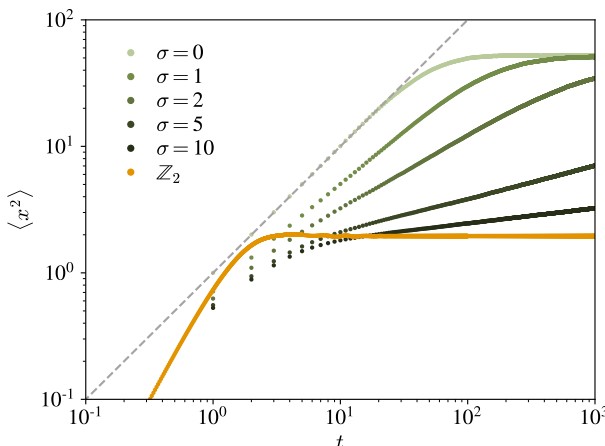

Figure 5: Squared displacement $\langle x^2 \rangle$ vs time $t$ for a spinon moving along a $\mathbb{Z}_2$ gauge chain (initialised in the middle of an $L = 25$ chain), for intermediate dephasing $\gamma = \Gamma/2$ where $\Gamma = 1$ is the hopping. This is contrasted to the behaviour of a classical 1D RW in presence of random onsite pinning energies, Gaussian distributed with zero mean and standard deviation $\sigma = 0, 1, 2, 5, 10$, at finite temperature $T = 1$. The data for the $\mathbb{Z}_2$ gauge chain are averaged over $10 \cdot 2^{10}$ random vison configurations, and those for the RW process are averaged over $10^4$ histories.

For weak disorder, we see a clear difference between the motion of a spinon in the presence or absence of visons. In the latter case, the spinon ballistically reaches the edges of the system ($\langle x^2 \rangle \simeq 50$ in our case). In the former, it saturates at a displacement of the order of one lattice spacing. The distinction survives up to very strong diagonal disorder. On the contrary, we see it becoming weaker and weaker as off-diagonal disorder becomes comparable to the spinon hopping energy scale, $w_{s,s'}^{(o)} \sim \Gamma \sim 1$. This is to be expected, and it will be a caveat to keep in mind when thinking about possible implementations on quantum simulator platforms.

We conclude that (at least up to off-diagonal disorder strengths of the order of half the hopping amplitude) the striking difference between the behaviour of spinons in presence or absence of visons can clearly be used as signature of quantum coherence, mutual semionic statistics and quantum spin liquid behaviour.

## 3.3 Classical diffusion

When attempting to find witnesses of any effects due to quantum coherence and interference, it is important to rule out possible competing classical (i.e., incoherent) processes that may lead to the same behaviour. In this case, one may be concerned about the limiting case where the spinons behave as incoherent stochastic particles random-walking (RW) on a strongly disordered 1D chain. The latter can in fact result in what looks like sub-diffusive or even 'localised' behaviour at intermediate times (whilst being a purely classical pheonmenon, devoid of any quantum coherence).

We compare the behaviour of our base system in presence of visons to a RW particle in 1D in presence of pinning disorder Gaussian distributed with zero mean and standard deviation $\sigma = 0, 1, 2, 5, 10$, where stochastic hopping is assumed to satisfy detailed balance with respect to the disorder and a reference temperature $T = 1$. The customary interpretation of Monte Carlo time as real time is assumed.

The results are shown in Fig. 5 and Fig. 6. The two behaviours are potentially distinguished by their short time regime $t \lesssim 1$ (in units of $1/\Gamma$ for the quantum chain, and in units of the Monte Carlo hopping time for the RW particle), which ought to be $\langle x^2 \rangle \sim t^2$ vs $\langle x^2 \rangle \sim t$

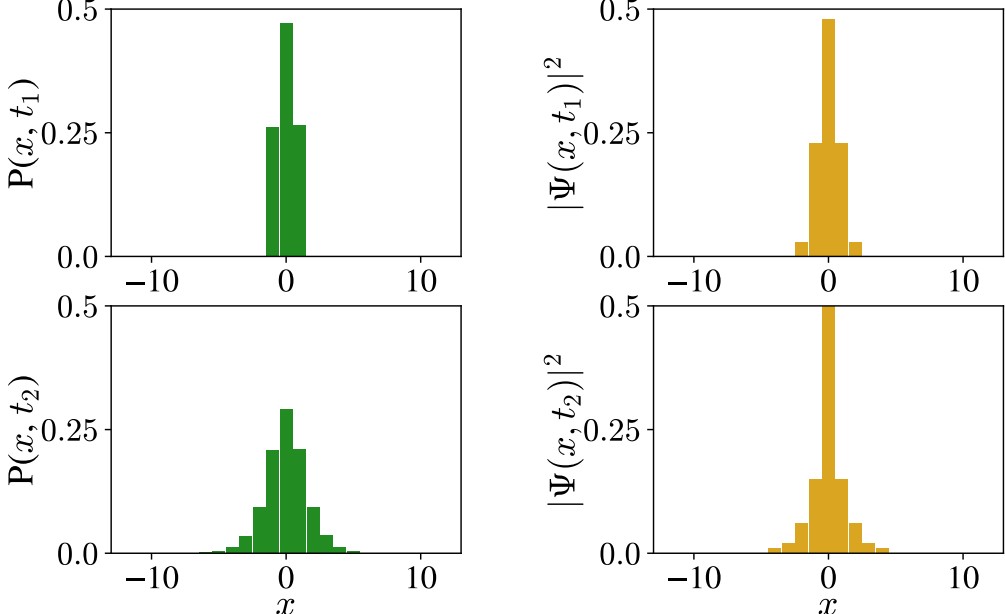

Figure 6: Probability distribution for a classical RW particle on a disordered 1D chain (left panels) in presence of strong disorder (Gaussian distributed with standard deviation $\sigma = 10$, $T = 1$), and $|\Psi(x,t)|^2$ for the spinon in our $\mathbb{Z}_2$ ladder (right panels), after initialising the particle/spinon in the centre of the system of size $L = 25$. Top panels: time $t_1 = 1$. Bottom panels: time $t_2 = 100$. (Here we make the working assumption that the characteristic inverse hopping time scale in the dissipative quantum system corresponds to the conventional unit time in Monte Carlo simulations of a particle performing a RW.) The RW probability distribution profile is averaged over $10^4$ histories and the spinon wavefunction $|\Psi(x,t)|^2$ for the $\mathbb{Z}_2$ ladder is averaged over $10 \cdot 2^{10}$ random vison configurations.

(quantum ballistic vs classical diffusive behaviour). However, accessing short times and short distances is often challenging and sometimes unreliable in quantum simulators and experiments; this is indeed the case in D-wave machines (see Sec. A).

At late times, it would be difficult to discern the two cases for, say, $\sigma \simeq 10$. However, one can elegantly resolve this dilemma by comparing the behaviour of the $\mathbb{Z}_2$ ladder with a modification thereof where the spins on the bottom leg are pinned by a large magnetic field. With these spins fixed, the system reduces to a 1D ferromagnetic Ising chain, and the spinon becomes a trivial domain wall (kink) therein. The toric code physics and the visons are killed in the process, and the behaviour in the base model reverts effectively to that of the $\mathbb{Z}_2$ ladder *without visons*, see the left panels in Fig. 3 and the blue curves for $\sigma_{0,1} = 0$ in Fig. 4. On the contrary, this change makes little difference to the RW particle on a disordered ladder; even if it is no longer able to hop onto the bottom sites of each plaquette, it will behave in a similar way. In summary, the difference between the behaviour of the $\mathbb{Z}_2$ ladder with and without a strong uniform pinning magnetic field on the bottom leg spins (see panel (b) in Fig. 1) provides a stark and reliable signature of quantum coherence and semionic statistics over classical RW motion.

### 3.4 Stroboscopic vison dynamics

So far we assumed the visons to be quasistatic on the time scales of the spinon motion. This is reasonable so long as transverse terms for the visons are negligibly small compared to the spinon hopping.[3]

Vison dynamics is however expected to play an important role at long times, when it is capable of (re)moving the cuts to spinon motion due to destructive mutual statistics interference. This allows spinons to travel to larger and larger distances, and the initially localised wave function will gradually spread out in a likely diffusive way (at least if the vison dynamics is stochastic).

It would therefore be interesting to see what effect vison motion might have on our results. Unfortunately, solving the full quantum mechanical system at finite temperature, including both spinon and vison dynamics, is beyond reach even for the relatively small systems and short time scales studied here.

In order to investigate this crossover from the initial localised behaviour (which we aim to use in order to probe $\mathbb{Z}_2$ quasiparticle behaviour and quantum phase coherence in the system) to the late time spreading due to vison motion, we devised the following toy model of stroboscopic evolution. We run simulations that alternate the quantum evolution of the spinon wave function for a time interval $\delta t$ with stochastic attempts to update the vison configuration at a single lattice site (i.e., updating the two adjacent bonds, as required by their emergent quasiparticle nature). Note that vison updates include both vison motion as well as (dissipative) pair creation and annihilation events.

For convenience, instead of performing the vison updates at regular time intervals $\delta t$, we perform them at random times drawn from a Poissonian distribution of characteristic time scale $\delta t$. The two approaches are equivalent at long times, and the latter avoids aliasing effects at short times caused by a regular stroboscopic pattern. We compute the spinon displacement after initialising it in the middle of a chain of length $L = 25$ by averaging over $2^{10}$ histories for the given $\delta t$ (starting from a random vison configuration of density $\rho_v = 1/2$).

For $\delta t \gg 1$ (recall that time is measured in units of $1/\Gamma$, with $\Gamma = 1$), we expect to recover the quasistatic limit used in the main text, and thence the results obtained so far; for $\delta t \ll 1$, we expect instead the vison configuration to thermalise well within the spinon dynamics time scales of interest in this work, and quantum diffusive behaviour ensues. The results of the stroboscopic vison dynamics model are illustrated in Fig. 7. As usual, we set the hopping amplitude $\Gamma = 1$, and correspondingly we set the unit of time ($\sim 1/\Gamma$). Within the remit of validity of this toy model, we see that the results in the main text hold for sufficiently slow stochastic vison dynamics ($\delta t \gtrsim 1$) – namely, so long as an intermediate-time localisation plateau around $\langle x^2 \rangle \sim 1$ is sufficiently prominent.

## 4 Connection between the microscopic CGS ladder and the effective $\mathbb{Z}_2$ ladder

In this section we connect the microscopic model, Hamiltonian Eq. (1), to the effective model, described by Eq. (3). Both models, depicted in Fig. 1(a) and 1(b), admit *exact* local $\mathbb{Z}_2$ gauge symmetries, the former via plaquette operators $G_p$ of Eq. (2) (as described in Ref. [20]), and the latter by operators $B_p$ that are the products $\sigma_\ell^x \sigma_{\ell'}^x$ of spin operators on the links $\ell$ and $\ell'$ bounding plaquette $p$. We also discuss the limit in which we can identify the low-energy

---

[3]As a matter of fact, they ought to be smaller than the vison energy cost, otherwise they can induce a phase transition out of the topological spin liquid phase where visons and spinons are well-defined quasiparticles with mutual semionic statistics.

quasiparticle excitations, spinons and visons, in the two models, and their mutual semionic statistics.

## 4.1 Derivation of the effective $\mathbb{Z}_2$ ladder Hamiltonian from the CGS ladder Hamiltonian

We begin by considering a single star of the CGS ladder, corresponding to a term labelled by $s$ in the first line of Eq. (1a). The spectrum of these isolated stars can be obtained exactly. The ground state has energy $E_0 = -4\sqrt{(2J)^2 + \Gamma_0^2}$ and it is 8-fold degenerate. These states have positive parity, i.e., the product of the four gauge spins on the legs of the star equals $P = +1$. The first excited state has energy $E_1 = -\sqrt{(4J)^2 + \Gamma_0^2} - 3|\Gamma_0|$ and it is also 8-fold degenerate, but with negative parity, $P = -1$. The second excited state is 32-fold degenerate, has $P = +1$, and energy $E_2 = -\sqrt{(4J)^2 + \Gamma_0^2} - |\Gamma_0|$. (Notice that for $\Gamma_0 = 0$ the first and second excited states become degenerate.)

Next we consider two coupled stars, as depicted in Fig. 1(c). We are particularly interested in the spinon gap and the spinon hopping amplitude. We pin the outer gauge spins to analyse separately the cases with even (0 or 2) and odd (1) number of spinons on the two-star system. For the even case, we pin all four gauge spins on the boundaries to $\sigma^z = +1$. For the odd case, we pin three to $\sigma^z = +1$ and the remaining one oppositely, $\sigma^z = -1$, as depicted in Fig. 1(c). We then diagonalise the system of 8 matter spins (4 in each star) and 4 gauge spins (2 on each of the upper and lower legs).

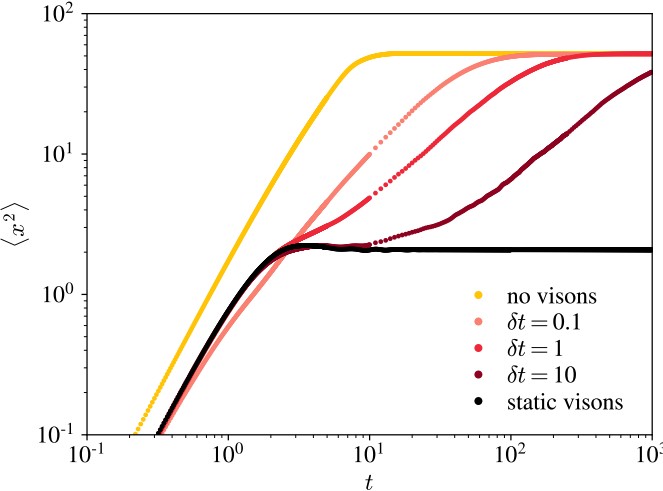

Figure 7: Behaviour of the spinon in our $\mathbb{Z}_2$ ladder in presence of stroboscopic stochastic vison dynamics with characteristic time scale $\delta t$ (as discussed in the main text). These results were obtained for chains of size $L = 25$, and time is expressed in units of $1/\Gamma$ ($\Gamma = 1$). The limit $\delta t \gg 1$ reproduces the earlier result for static visons (black curve). The case without mutual semionic statistics (equivalently, no visons in the system) is also shown for reference (yellow curve). The system crosses over to trivially diffusive behavior as $\delta t$ is reduced (shown by curves in progressively lighter shades of red). One can approximately identify a threshold around $\delta t \sim 1$ below which our proposed signature of the mutual semionic statistics becomes no longer viable (middle curve in the figure). The results for dynamical visons are averaged over $2^{10}$ histories with characteristic timescale $\delta t$ (starting from a random vison configuration of density $\rho_v = 1/2$). For the static visons, the data are averaged over $2^{10}$ random vison configurations of density $\rho_v = 1/2$.

In Fig. 8 we show the spectrum for the two-star assembly for the CGS ladder case (left) and compare it to the equivalent two-star assembly for the $\mathbb{Z}_2$ ladder case (right). We show the first 8 low energy states for each case. We fix the couplings $J = 1$ in the CGS ladder. The top panel shows the spectrum for $\Gamma_0 = 0$. (Notice that the first and second excited states become degenerate in the CGS system when the transverse field $\Gamma_0$ is switched off, as discussed above.) The labels on the levels correspond to the signs of $\sigma^z$ for the mobile gauge spins on the upper and lower links [coloured in orange in Fig. 1(c)]. In the region labeled as even ($P = +1$), there are no spinon defects in the case of the lower energy levels, and there are two spinons – one on each of the two stars – in the case of the higher energy levels. The two-fold degeneracy of these states corresponds to the choices of signs of the gauge spins on the upper and lower links. In the region labeled as odd, there is only one spinon, which can sit in one star or the

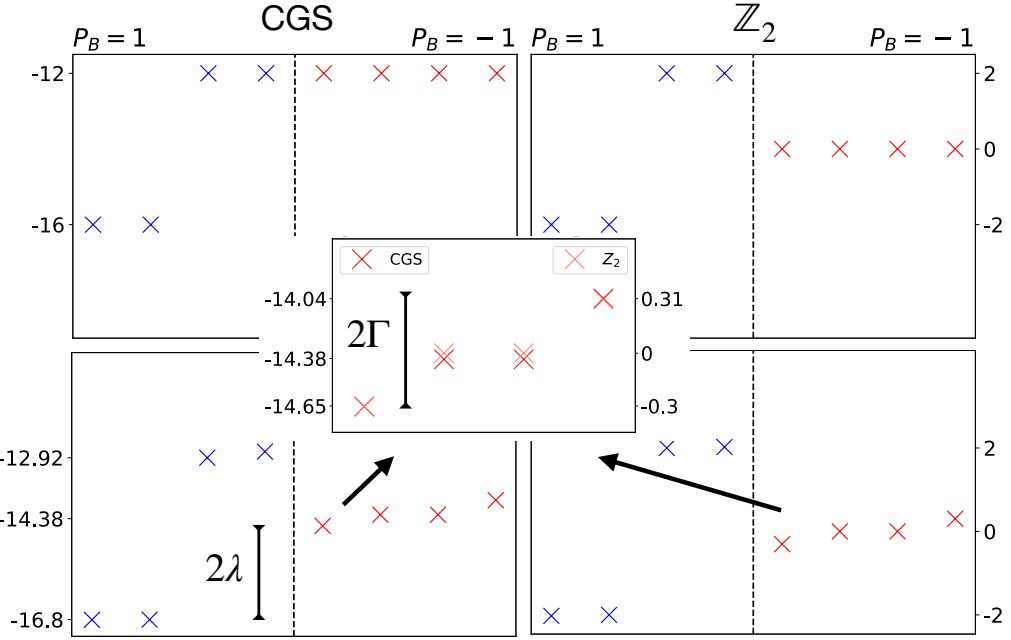

Figure 8: Energy spectrum of two coupled stars of the CGS and $\mathbb{Z}_2$ ladder, with fixed couplings $J = 1$ for the CGS and $\lambda = 1$ for the $\mathbb{Z}_2$ ladder. Only the lowest 8 energies are shown for each panel, and the vertical dashed lines separate the cases of even (blue) and odd (red) number of spinons. In top two panels, the transverse fields are set to zero, $\Gamma_0 = 0 = \Gamma/2$. The ground states for both ladders are 2-fold degenerate, with no spinons and the two coupled stars in their lowest energy state, $E_{\text{star}}^{\text{CGS}} = -8$ and $E_{\text{star}}^{\mathbb{Z}_2} = -1$. The first and second excited states correspond to single spinon excitation on either of the stars and two-spinon configuration respectively. Notice that the first and second excited states of CGS ladder are exactly degenerate when $\Gamma_0 = 0$ as described in the main text. Transverse fields are turned on in the bottom two panels, $\Gamma_0 = 0.6$ for the CGS case and $\Gamma/2 = 0.1525$ for the $\mathbb{Z}_2$ case. In the odd case ($P_B = -1$), the four degenerate first excited states split into three different energies, and the difference between the lowest and the highest, associated to the spinon hopping, is exactly $2\Gamma$ in the $\mathbb{Z}_2$ ladder, labelled in the inset. The inset shows that the hopping energy scales are the same for CGS and $\mathbb{Z}_2$ ladders when $\Gamma_0 = 0.6$ and $\Gamma/2 = 0.1525$ respectively. The spinon gap, defined as the difference between the ground state energy and first excited state (specifically the gap to the 2 middle degenerate states), is exactly $2\lambda$ in the $\mathbb{Z}_2$ case, labelled in the lower left panel.

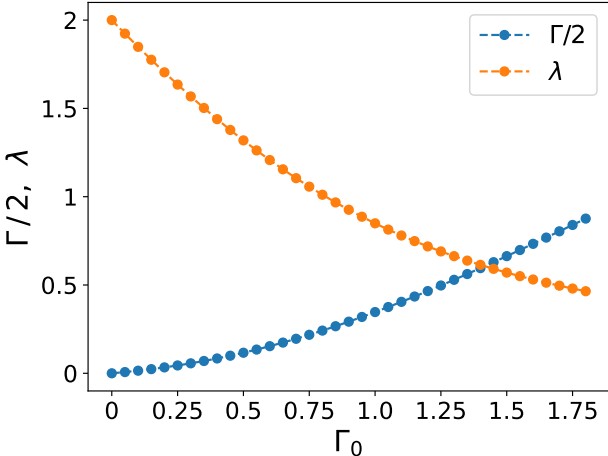

Figure 9: Effective couplings $\lambda$ (orange) and $\Gamma/2$ (blue) of the $\mathbb{Z}_2$ ladder as a function of $\Gamma_0$ of CGS ladder when $J = 1$. The transverse field parameter $\Gamma/2$ defines the spinon hopping energy scale, $\lambda$ sets the spinon gap in $\mathbb{Z}_2$ ladder, and both can be obtained from analysing the energy spectrum of CGS ladder, as shown in Fig. 8. As $\Gamma_0$ increases, $\Gamma/2$ increases and $\lambda$ decreases, and the two cross around the point when $\Gamma_0 = 1.4$.

other. There are four such states, again labeled by the signs of $\sigma^z$ for the gauge spins on the upper and lower links. The spectrum for the $\mathbb{Z}_2$ ladder case is shown on the right panel for comparison.

The lower panel of Fig. 8 shows the spectrum once the transverse field $\Gamma_0$ is turned on ($\Gamma_0 = 0.6$) in the CGS system. On the right we show the spectrum for the equivalent two-star segment of the $\mathbb{Z}_2$ ladder system for $\lambda = 1$ and $\Gamma/2 = 0.1525$. In both the CGS ladder and the $\mathbb{Z}_2$ ladder cases, the spinons can hop between the two stars due to the transverse fields, which results in the energy level splitting shown on the side labeled odd, for both cases. We show in the inset a magnified window in which the splittings can be better observed. Notice that the four states in the odd case (which are degenerate when $\Gamma_0 = 0 = \Gamma/2$) split into three different energies with degeneracies 1, 2, and 1, respectively. These states (and degeneracies) are understood as follows. The operators $G_p$ in the case of the CGS ladder and $B_p$ in that of the $\mathbb{Z}_2$ ladder commute with the Hamiltonians Eq. (1a) and Eq. (3), respectively. The eigenvalues of these operators relate to the presence or absence of a vison within the plaquette. In the absence of a vison, the spinon can hop between the two stars, leading to a symmetric and anti-symmetric splitting. In the presence of a vison, the hopping is switched off (due to perfect destructive interference), and the two levels remain exactly degenerate. The energy splittings among these four states, homologous in the CGS ladder and in the $\mathbb{Z}_2$ ladder, allow us to read off the effective couplings.

Explicitly, the couplings $\lambda$ and $\Gamma/2$ of Eq. 3 can be directly extracted from the CGS energy spectrum, where the energy difference between the lowest and the highest of the splittings of first excited states is $2\Gamma$, and the spinon gap, defined as the difference between the ground state energy and first excited state (specifically the gap to the 2 middle degenerate states), is exactly $2\lambda$ in the $\mathbb{Z}_2$ case, shown in Fig, 8. Fig. 9 shows the dependence of the couplings $\lambda$ and $\Gamma/2$ on the applied transverse field $\Gamma_0$ in the CGS ladder. Near $\Gamma_0 = 1.4$, $\lambda$ and $\Gamma/2$ become comparable. (We remark that when $\lambda < \Gamma/2$ the state with a spinon becomes the ground state, signalling the onset of spinon condensation; the precise ratio $\Gamma/2\lambda$ for the phase transition, however, cannot be extracted from the simple two-star calculation.)

## 4.2 Analytic connection between CGS and $\mathbb{Z}_2$ ladders in a perturbative limit

In Sec. 4.1, we provided a mapping between the parameters of the CGS and $\mathbb{Z}_2$ ladders in Fig. 9. Here, we provide an exact correspondence between the two models in the limit where the "$ZZ$" coupling between gauge and matter spins is the dominant energy scale. While the arguments presented in this section are perturbative in nature, the correspondence is expected to hold at a qualitative level in parameter regimes where perturbation theory is not quantitatively accurate.

For concreteness, consider Eq. (1a) with separate magnetic field strengths for the matter and gauge spins, $\Gamma_m$ and $\Gamma_g$, respectively:

$$H = -\sum_s \left[ J \sum_{\substack{a \in s \\ \ell \in s}} W_{a\ell} \, \mu_a^z \, \sigma_\ell^z + \Gamma_m \sum_{a \in s} \mu_a^x \right] - \Gamma_g \sum_\ell \sigma_\ell^x.$$

We will be concerned with the regime in which $|J| \gg |\Gamma_m| \gg |\Gamma_g|$. To analyze the behavior of the model in this regime, let us first forget about the separation of scales between $\Gamma_m$ and $\Gamma_g$, and just perform (degenerate) perturbation theory assuming that $J$ is positive and large compared to both transverse fields.

First, let us consider a single star consisting of both matter and gauge spins in the absence of any transverse fields, $\Gamma_m = \Gamma_g = 0$, i.e., the Hamiltonian is entirely commuting and consists of the Hadamard matrix (1b) coupling the $z$ components of the gauge and matter spins. For a given configuration of gauge spins, the matter spins can either be aligned or antialigned with the effective local (gauge) field acting on them. If all matter spins are aligned with their local fields, the ground state energy of star satisfies

$$E_0 = -2J(3 + A_s). \tag{7}$$

Hence, parity-even gauge-spin configurations ($A_s = 1$) minimize the energy of the star: $E_0 = -8J$. For such-parity even configuration, there exists a *unique* matter-spin configuration (i.e., none of the local fields acting on the matter spins vanish). Hence, the star has eight degenerate ground state configurations corresponding to parity-even configurations of gauge spins.

Next, consider the excited states of the star. There are two possibilities: (i) the gauge spins are parity odd, $A_s = -1$, or (ii) the gauge spins are parity even, but at least one matter spin is not aligned with its local field. These two types of defects happen to be degenerate with $E = -4J$. Case (i), corresponding to 'defective' gauge spins, will be referred to as a 'spinon' excitation. Since there does not exist a local operator that can globally fix the parity of an isolated parity-odd star, there will not exist matrix elements connecting the two species of excitations in (i) and (ii); the decay of a spinon into a defective matter spin will therefore not occur within our perturbative expansion. For each parity-odd gauge spin configuration, there are *eight* degenerate matter spin configurations, since the local field vanishes on three of the four matter spins. This can be illustrated schematically in the following example:

$$\begin{array}{ccccc} & & \textcolor{red}{+} & & \\ & & \textcolor{blue}{?} & & \\ \textcolor{red}{+} & \textcolor{blue}{?} & \textcolor{red}{+} & \textcolor{blue}{?} & \textcolor{red}{+} \\ & & \textcolor{blue}{+} & & \\ & & \textcolor{red}{-} & & \end{array}$$

where the red (gauge) spins form an $A_s = -1$ configuration, and the blue (matter) spins are in the corresponding lowest-energy configuration, which is not uniquely specified (the matter spins represented by question marks can either take the value $+$ or $-$, without changing the energy of the system).

We now consider connecting the stars together in the manner shown in Fig. 1. Furthermore, we turn on the magnetic fields acting on both gauge and matter spins. The combinatorial gauge symmetry (2) remains *exact* in the presence of such magnetic fields; we therefore work with a set of states that diagonalize the $G_p$ operators. The eigenvalues $G_p = \pm 1$ determine the locations of the 'vison' excitations. The magnetic field $\Gamma_m$ induces transitions between the eight degenerate matter spin configurations on each star, while $\Gamma_g$ introduces matrix elements between adjacent star configurations. To make connections with the results presented in the main text, let us first construct the effective Hamiltonian for the two-star system from Sec. 4.1, within first-order degenerate perturbation theory. One finds that the spinon (whose presence can be enforced with appropriate boundary conditions, as discussed in Sec. 4.1) hops on the following effective lattice:

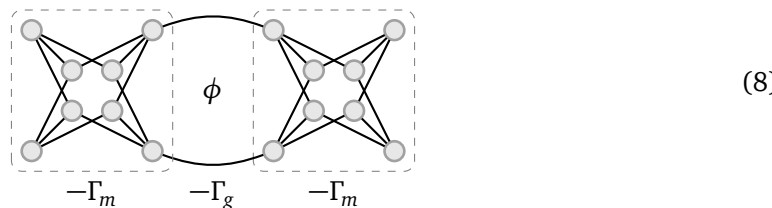

(8)

Each vertex in the graph represents a symmetrized state $\propto (1 \pm G_p)|\sigma, \mu\rangle$, where the sign $\pm$ determines the absence or presence of a vison, while each edge corresponds to a connection between these states induced by a transverse field. The dashed gray boxes indicate the 'internal' sites belonging to each star. The $\Gamma_m$ transverse field induces intra-star hopping, while $\Gamma_g$ leads to inter-star hopping. If $G_p = 1$, the spinon hops on the effective lattice with vanishing flux threading the central loop, $\phi = 0$. Conversely, in the $G_p = -1$ sector, the presence of the vison leads to $\phi = \pi$ threading the central loop.

If $\Gamma_g = 0$, the two stars remain completely decoupled, and there exist eight 'internal' states associated to each star. Let the ground and first-excited states supported on star $s$ be denoted by $|\varphi_0(s)\rangle$ and $|\varphi_1(s)\rangle$, with energies $\epsilon_0$ and $\epsilon_1$, respectively. If $\Gamma_g$ is turned on, and $|\Gamma_g|$ is much smaller than the gap between the ground and first-excited internal state of an isolated star, $\epsilon_1 - \epsilon_0$ (equal to $2\Gamma_m$), then we can project the dynamics into the lowest band. This results in the effective Hamiltonian

$$H_{\text{eff}} = -\Gamma_g |M|^2 [1 + e^{i\phi}](|\varphi_0(s)\rangle \langle\varphi_0(s+1)| + \text{h.c.}), \qquad (9)$$

where $|M| = |\langle 0|\varphi_0\rangle|$ is the projection of the local ground state onto one of the outermost sites. In our system, $\phi = \pi$ and we obtain perfect destructive interference: the spinon is unable to hop and remains exactly localized on one of the two stars. Extending the above Hamiltonian to a one-dimensional chain of stars by including a summation over $s$ justifies quantitatively the hopping Hamiltonian presented in Eq. (5) in Sec. 3.1.

If higher-order contributions in $\Gamma_g$ are taken into account, perfect destructive interference is lost, but the spinon remains *approximately* localized on one site or the other, since the residual hopping terms between the two stars are second order in $\Gamma_g$, namely $O(\Gamma_g^2)$. Numerically, we find that this behavior actually persists all the way up to equal transverse fields $\Gamma_g = \Gamma_m$, where, for the most localized eigenstates,[4] there is only approximately a 4% chance of finding the spinon on the other star. The quantitative effect of residual hopping between adjacent stars – removing perfect destructive interference – was studied in Sec. 3.2 by adding off-diagonal disorder to the effective hopping Hamiltonian.

Finally, we note that this perturbative mapping also quantitatively explains the small-$\Gamma$ behavior of Fig. 9. Diagonalizing the 16-dimensional Hamiltonian corresponding to the lattice

---

[4]When the vison is present, the ground state is two-fold degenerate. Hence, we find the "most localized" linear combination of ground states by extremizing the probability of finding the spinon on the left vs the right star.

in (8), we find that the parameters $\lambda$ and $\Gamma/2$, which set the spinon gap and the spinon hopping energy scale, respectively, are given by

$$\lambda = 2J - \frac{3.098}{2}\Gamma_0 + O(\Gamma_0^2), \tag{10a}$$

$$\frac{1}{2}\Gamma = \frac{0.515}{4}\Gamma_0 + O(\Gamma_0^2), \tag{10b}$$

where $\Gamma_0 = \Gamma_g = \Gamma_m$.

To conclude, we have shown that, for sufficiently small magnetic fields, the CGS ladder (1) can be understood as a tight-binding model for spinons. The lattice geometry derives from the Hadamard matrix (1b), which couples matter and gauge spins, and involves eight states per star. The exact combinatorial gauge symmetry implies that visons ($G_p = -1$) remain exactly static, and are equivalent to $\pi$ fluxes threading the lattice on which the spinons hop. This perturbative mapping is valid irrespective of the relative strengths of the two magnetic fields. However, to obtain *perfect* destructive interference, leading to compact-localized spinon eigenstates, it is necessary to consider the case where the transverse field on gauge spins is much weaker than the transverse field for matter spins, see Eq. (9). In this case, projecting the dynamics into the lowest band is well-controlled and we obtain an effective tight-binding model with one state per site. This is the model (5) we considered in Sec. 3.1.

## 5 Testing quantumness

In this manuscript, we referred several times to the possibility of using our results to test the quantum coherence present in a programmable platform or $\mathbb{Z}_2$ quantum spin liquid candidate system. Before closing, it is worth summarising our thoughts in a dedicated section.

Detecting quantum coherence in systems where this is present over long time and length scales is generally not a challenge. For instance, the time evolution of a particle in 1D exhibits characteristic quantum oscillatory behaviour; and the energy eigenfunctions of a particle in a box easily betray their wave-like nature by looking at the probability density [29].

This is however not the case we were referring to in our work. What we had in mind are for example quantum spin liquid candidate materials at temperatures where coherence operates at most over few lattice spacings; or, similarly, quantum simulators where some coherence is expected, but of limited range and unclear relevance. In such settings, one would like to ask: What is the 'least amount' of phase coherence that can be meaningfully detected? Namely, the least quantum coherence that leads to a distinct and measurable change in behaviour of the system, which would disappear if this coherence is reduced any further?

Quantum phenomena and measurements that may allow us to answer these questions are few and far between. With little coherence in a system, its effects also become progressively more subtle and hard to detect with certainty in a noisy environment with finite experimental accuracy or temporal resolution. It is in this context that our results stand out as a notable exception: the presence or absence of quantum phase coherence around an elementary plaquette of the system gives rise to the presence or absence of constructive / destructive interference of a spinon wavefunction, depending on the vison occupation number of the plaquette due to their mutual semionic statistics. As we have shown in our work, this semionic interference has a remarkably large effect on the collective behaviour of the system, and the effect is robust to disorder, noise and dephasing. Moreover, we have shown how it can be distinguished from potentially similar classical stochastic behaviour (e.g., random walk with strong pinning disorder) by contrasting the behaviour with and without an applied field on the lower leg of the $\mathbb{Z}_2$ ladder). We therefore argue that our work provides a clear and distinctive signature of quantum mechanical behaviour in the system at the single-plaquette level – arguably the

lowest level of coherence below which the system would be described quantitatively well by classical statistical mechanical modelling, and arguably the least phase coherence needed to observe any collective quantum mechanical behaviour. Hence the wording we used in the manuscript, of our work providing a way to test quantumness in these systems.

# 6   Conclusion

In this work, we considered the implementation of topological quantum spin liquid phases – specifically, $\mathbb{Z}_2$ spin liquids – in programmable quantum devices, using combinatorial gauge symmetry. We focused our attention on probing such exotic, highly entangled states and on devising smoking-gun measurements that enable us to determine their presence, using the fractional statistics of their topological excitations. We designed and studied quasi-1D architectures where we demonstrate that quasiparticle interferometry leads to distinct signatures of the underlying spin liquid behaviour. We tested the robustness of these effects against disorder and dephasing, which are expected to be present in noisy quantum programmable devices (and in experimental settings in general).

Our results show how spinon transport along a ladder is curtailed by destructive interference in presence of visons. Contrasting this behaviour to the motion of the spinon when one leg of the ladder is pinned by a strong applied field – thus removing interference effects – produces a distinct signature of the non-trivial mutual statistics.

While on the one hand our design and results can be used to establish the presence of these exotic topological phases of matter, they can also conversely be used to test the 'quantumness' of the platform on which the quantum spin liquid is realised. Indeed, the signature that we propose relies on exquisitely quantum interference effects (due to fractional statistics) that are spoilt once enough dephasing prevents the wave function of a spinon to interfere with itself after propagating on the lattice along different paths.

In Appendix A, we investigate the possible implementation of our scheme on a D-wave device. Our results show that the DW-2000Q simulator lies just outside the low temperature limit where topological phase coherence effects become observable in the form of destructive interference due to mutual statistics. Foreseeable improvements might bring this architecture within reach of implementing quantum spin liquid phases in the near future.

Several other platforms have been made available by recent advances in programmable quantum devices where one could repeat our study, such as QuEra [30], a Rydberg-atom based computing platform. Some of them may already be in the parameter regime where our non-trivial signature of fractional statistics can be observed.

In this work we focused on quasi-1D systems as they provide some of the simplest implementations, largest systems (by embedding a meandering 1D line in 2D), and clearest signatures of quantum coherence and fractional statistics. However, the topological nature of the quasiparticle excitations in quantum spin liquids leads to important signatures also in 2D, as already pointed out in Refs. [16,17]. Further work in this direction is ongoing but it is beyond the scope of the present paper.

For our purpose, we considered it sufficient to model the vison dynamics as a classical stochastic process. However, more subtle effects could derive from the interplay of quantum vison and spinon dynamics. Their investigation is a significantly taller order and is an interesting direction for future work.

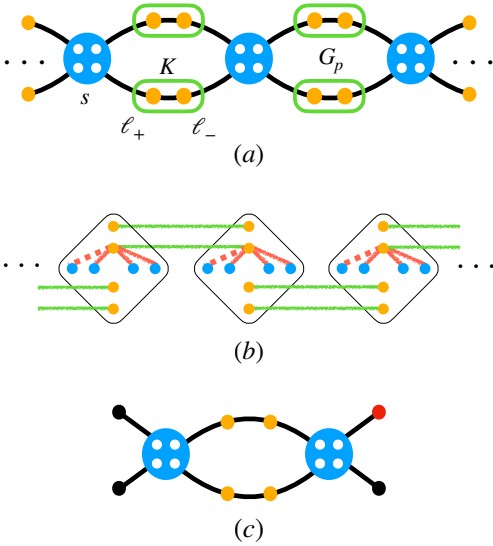

Figure 10: Diagrams of CGS ladder with additional $K$-couplings and its embedding onto the D-Wave chimera architecture. (a) Geometry of the CGS ladder with two gauge spins (orange dots) at $\ell_\pm$ on each link $\ell$, and four matter spins (white dots) on each vertex (large blue discs) at site $s$. A strong ferromagnetic coupling of strength $K$ forces the alignment of the two gauge spins that occupy the same link. The local $\mathbb{Z}_2$ symmetry is generated by $G_p$, that both flips the four gauge spins around a plaquette, and flips and permutes the states of the matter spins on the two stars neighbouring the plaquette. (b) The embedding of the geometry in (a) to the D-Wave DW-2000Q chimera architecture. Each unit cell (of 8 spins) embeds a star operator of the CGS ladder in which the horizontal spins (blue dots) are the matter spins, and the vertical spins (orange dots) are the gauge spins. Two adjacent stars are coupled by connecting the corresponding gauge spins with strong ferromagnetic fields (green lines). (c) Example of an isolated pair of coupled stars of the CGS ladder with the $K$-coupling, used to extract the effective spinon cost and hopping amplitude from the energy splittings in the spectrum. The boundary gauge spins are pinned to be three up (black dots) and one down (red dot) so the ground state of this configuration has exactly one spinon.

# Acknowledgments

We are grateful to Dmitry Green, Andrew King, Mohammad Amin, Richard Harris, and Jack Raymond for insightful discussions.

**Funding information** This work was partly supported by the Engineering and Physical Sciences Research Council (EPSRC) Grants No. EP/K028960/1, No. EP/P034616/1, and No. EP/T028580/1 (C.Ca.), by the U.S. Department of Energy (DOE), Office of Science, Basic Energy Sciences (BES) under Award #DE-SC0021346 (O.H.), and by the DOE Grant No. DE-FG02-06ER46316 (S.Z. and C.Ch.).

# A  D-Wave embedding and implementation

The results presented in the main text show that: (i) it is possible to realise a CGS ladder with an *exact* local $\mathbb{Z}_2$ gauge symmetry using Hamiltonian Eq. (1) containing only physical 1- and 2-spin interactions; and (ii) there is a window of parameters, e.g., disorder strength and dissipation rates, within which it is possible to observe qualitative differences in behaviour that reflect the mutual statistics of spinons and visons. In this Appendix we show our proposed experiment can be performed in a D-Wave device – a quantum annealing machine that operates under the following Hamiltonian,

$$H = \frac{A(s)}{2} \sum_i \sigma_i^x + \frac{B(s)}{2} \left[ \sum_{i,j} J_{ij} \sigma_i^z \sigma_j^z + \sum_i h_i \sigma_i^z \right] , \tag{A.1}$$

where $J_{ij}$ and $h_i$ are the programmable $ZZ$-couplings and longitudinal fields respectively. We argue that Hamiltonian Eq. (1) can be implemented in a DW-2000Q device with an extra gadget,

$$H = -\sum_s \left[ J \sum_{\substack{a \in s \\ i \in s}} W_{ai}\, \sigma_i^z \mu_a^z + \Gamma_0 \sum_{a \in s} \mu_a^x \right] - \sum_\ell \left[ K\, \sigma_{\ell_-}^z\, \sigma_{\ell_+}^z + \Gamma_0 \sum_{i \in \ell} \sigma_i^x \right], \tag{A.2}$$

where a ferromagnetic term $K$ favours the alignment of two neighbouring gauge spins $\ell_-$ and $\ell_+$ on a link $\ell$, to realise a single effective gauge spin, as depicted in Fig. 10(a). This CGS ladder with $K$-couplings also possesses the combinatorial gauge symmetry that gives rise to the $\mathbb{Z}_2$ spin liquid phase, and it can also be mapped to the same quasi-1D toric code with Hamiltonian Eq. (3), depicted in Fig. 1(b).

We proceed to extract the connection between the couplings of Hamiltonian Eq. (A.2) and those of the effective model with Hamiltonian Eq. (3), similarly to the study in Section 4. We isolate two CGS stars and couple them via the second line of Eq. (A.2), and study the low energy spectrum of this system. The outer gauge spins are pinned in order to analyse separately the cases of odd and even number of spinons; the odd case is depicted in Fig. 10(d).

Fig. 11 shows the lowest 8 energy states for both even (0 or 2) and odd (1) number of spinons, separated by vertical dashed lines. The spectrum of the two-star assembly of the CGS ladder with $K$-couplings is shown on the left panels, and that of the equivalent two-star assembly of the $\mathbb{Z}_2$ ladder is shown on the right ones. We fix the couplings $J = 1$ and $K = 3$ in the CGS system, and $\lambda = 1$ in the $\mathbb{Z}_2$ ladder. The layout of energy spectrum for CGS ladder with $K$-couplings is identical to that of Fig. 8 in Section 4, in which the ground states are two-fold degenerate, and the first and second excited states are degenerate when $\Gamma_0 = 0$ (Fig. 11 top left panel). Notice that there is a universal down-shift of energy levels of the CGS ladder with $K$-couplings by 6 (in units of $J$) due to the ferromagnetic interactions of two gauge spins on the links with coupling strength $K = 3$. The spinon hopping amplitude can be extracted directly from the level splittings of the first excited states (corresponding to the single spinon case) when the transverse field is turned on, $\Gamma_0 = 0.6$ in the CGS system and $\Gamma/2 = 0.0375$ for $\mathbb{Z}_2$ ladder, shown in Fig. 11 lower panels. The inset shows a magnified window in which the splittings can be better observed. (We note that the the splitting is suppressed as compared to the case without $K$-couplings of Section 4.)

Explicitly, we obtain the couplings of Eq. (3) from the energy scales for the spinon gap, $\Delta_s$, and the spinon hopping energy scale, $\Delta_h$, labeled in Fig. 11. Fig. 12 shows the dependence of the energy scales $\Delta_h$ and $\Delta_s$ on the applied transverse field $\Gamma_0$ in the CGS ladder with

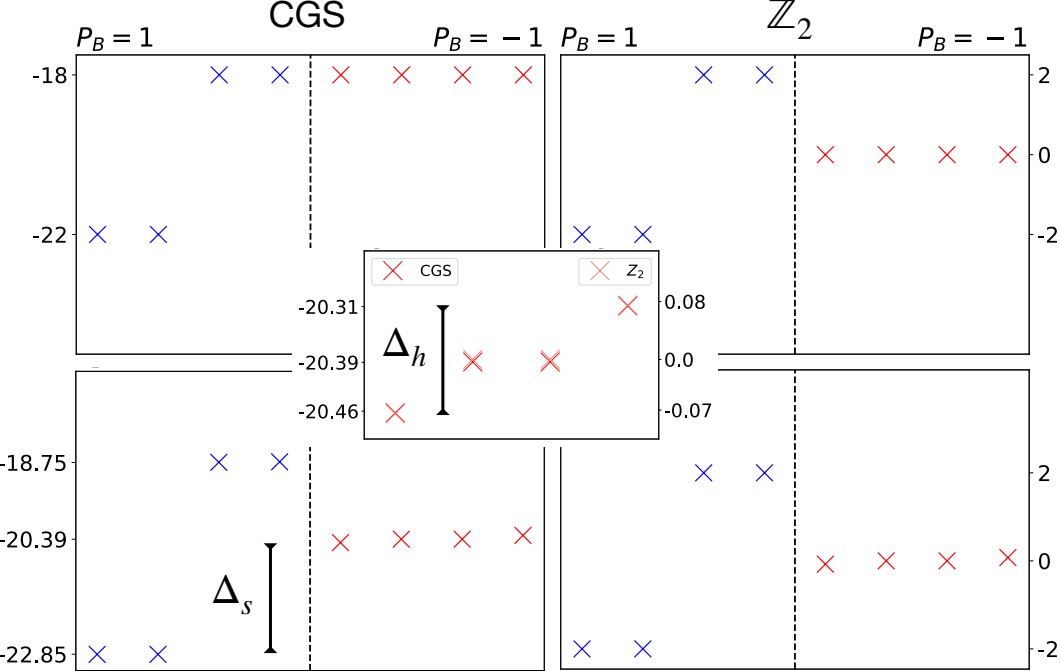

Figure 11: Energy spectrum comparison between two coupled stars of the CGS ladder with $K$-couplings and the $\mathbb{Z}_2$ ladder. The couplings $J = 1$ and $K = 3$ are fixed for the CGS ladder and $\lambda = 1$ for the $\mathbb{Z}_2$ one. Only the 8 lowest energy levels are shown for each panel, and the vertical dashed lines separate the cases of even (coloured in blue and labelled as $P = 1$) and odd (coloured in red and labelled as $P = -1$) number of spinons. In top two panels, transverse fields are zero, $\Gamma_0 = 0 = \Gamma/2$, and the ground state energy for the CGS case is $E_0^{\mathrm{CGS}} = -22$, where an energy $-6$ comes from two $K$-couplings connecting the gauge spins on the two links. Transverse fields are turned on in the bottom two panels, $\Gamma_0 = 0.6$ for the CGS with $K$-couplings and $\Gamma/2 = 0.0375$ for the $\mathbb{Z}_2$ ladder. In the odd case, the four degenerate states of the first excited state split, and the hopping energy scale is the difference between the lowest and highest states, labelled as $\Delta_h$ in the inset. The hopping energy scale is 0.15 when $\Gamma_0 = 0.6$ and $\Gamma/2 = 0.0375$, where the splitting in the $\mathbb{Z}_2$ ladder is exactly $2\Gamma$. Notice that, compared to Fig. 8 in Section 4, the coupling $K$ suppresses the splittings. The spinon gap $\Delta_s$ is defined as the difference between the ground state and the first excited states (specifically, the 2 middle degenerate states), as shown in the lower left panel.

$K$-couplings. The couplings $\lambda$ and $\Gamma/2$ of the $\mathbb{Z}_2$ ladder can be read off from

$$\lambda = \Delta_s/2\,,$$
$$\Gamma/2 = \Delta_h/4\,.$$

In particular, near the crossing where $\Delta_s$ and $\Delta_h$ are comparable, at $\Gamma_0 = 1.45$, we obtain $\lambda = 0.67$ and $\Gamma/2 = 0.31$.

The DW-2000Q device has $16 \times 16$ unit cells with 8 qubits in each, in total 2048 qubits. Its qubit-qubit connectivity allows the CGS ladder shown in Fig. 10(a), with Hamiltonian Eq.(A.2), to be implemented via the couplings shown in Fig. 10(b). The proposed experiment can be performed using the reverse annealing protocol in a D-Wave device. An initial state with a single spinon can be realised by setting the values of relevant qubits to be $\pm 1$ such that star operators are in their $A_s = 1$ eigenvalues on all vertices except on the vertex where

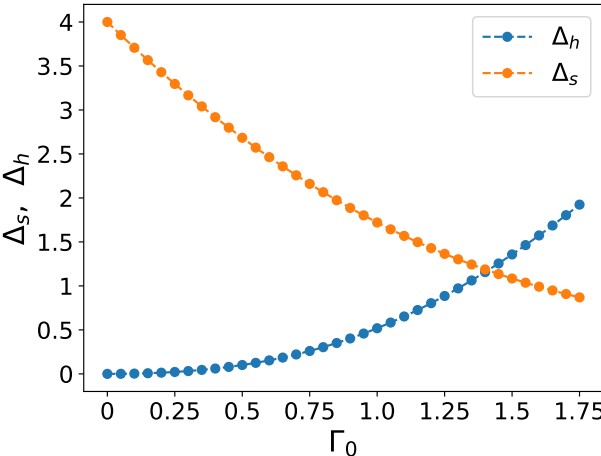

Figure 12: Spinon energy gap $\Delta_s$ (in orange) and spinon hopping gap $\Delta_h$ (in blue) of the CGS model with $K$-couplings as a function of $\Gamma_0$ when $J = 1$ and $K = 3$.

spinon is located, on which $A_s = -1$. The hopping can be activated by switching $\Gamma_0$ on for a certain amount of time, $\tau$. Measurements in D-Wave devices can only be done at zero transverse fields. We then quickly switch off $\Gamma_0$ back to zero and measure each qubit in its $z$-basis. The position to which the spinon travels can be read off the final configuration, by finding which vertex has $A_s = -1$. The set of final configurations in such an experiment allows one to extract the probability distribution for the final position of the spinon. Throughout this protocol one must keep $\Gamma_0$ below a threshold value so that multiple spinons are not generated, and instead only the single spinon imprinted in the initial condition is present at all times. We can set the boundary condition in a way to allow only odd number of spinons in the ladder to eliminate the case where extra spinons are created during the experiment. The configurations with two spinons are not allowed by the boundary condition and three spinons are energetically less probable.

In D-Wave devices, quantum annealing is realised by adiabatically changing the values of the parameters $A(s)$ and $B(s)$ in Eq. (A.1), controlled by the variable $s$. The schedule to change $A(s)$ and $B(s)$ is fixed in D-Wave devices, see [31]. In Eq. (A.1), the coupling strength of the Ising part [controlled by $B(s)$] increases, and the transverse field strength [controlled by $A(s)$] decreases, as $s$ increases from 0 to 1. The reverse annealing protocol allows one to start the experiment at the $s = 1$ endpoint and decrease $s$ to turn on the transverse field while the coupling strength in the Ising part is decreased.

The proposed set of parameters for our experiment in DW-2000Q is as follows. One can use $J^{\text{DWave}} = 1$, $K^{\text{DWave}} = 3$ at $s = 1$ limit, which corresponds to physical values $J = 2.43\,\text{GHz}$, $K = 7.27\,\text{GHz}$.[5] The transverse field should be turned on as quickly as possible to a programmable value $\Gamma^{\text{DWave}}$, which corresponds to decrease $s$. The base temperature where D-Wave operates is around 13 mK, or 0.27 GHz. In the reverse-annealing protocol, when the transverse field $\Gamma_0$ is increased, the physical values of $J$ and $K$ are simultaneously decreased. For instance, for $\Gamma_0^{\text{DWave}} = 0.61$, which corresponds to a physical value $\Gamma_0 = 0.65\,\text{GHz}$, $J = 0.46\,\text{GHz}$.

These experimental parameters can be used to obtain the effective $\mathbb{Z}_2$ ladder coupling $2\Gamma$, as discussed above. For $\Gamma_0/J \sim 1.45$ and $K/J \sim 3$, we find that $\Gamma/2J \sim 0.31$ (this is the case for which the spinon hopping energy scale is right below the spinon gap, see Fig. 12), corresponding to $\Gamma/2 \sim 0.14\,\text{GHz}$. This value is about 50% less than the operating temperature, posing a first limitation in carrying out the experiment in a D-Wave-2000Q device. Second,

---

[5]When $K = 3J$ and $\Gamma_0 = 0$, the energy for the spinon to reside on a link or on a star coincide.

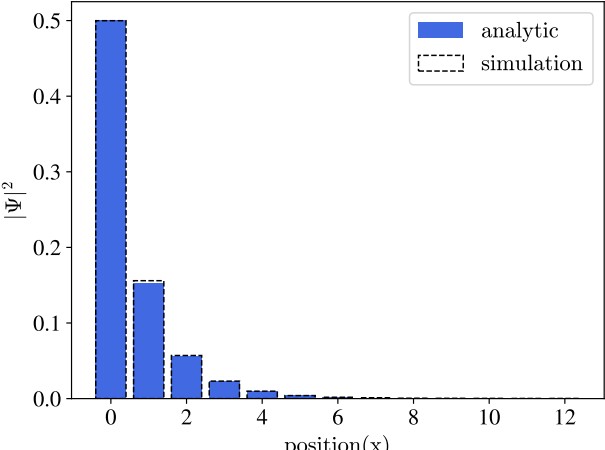

Figure 13: Comparison between the analytical result (blue bars) and the simulated wave function (dashed bars) in the long time limit for the particle on the $\mathbb{Z}_2$ chain coupled to a thermal bath with dephasing rate $\gamma = \Gamma/2$, where $\Gamma = 1$ is a transverse field. The numerical data are averaged over 5120 vison configurations.

the associated time scale is $\tau_\Gamma = (\Gamma/2)^{-1} \sim 7.14\,\text{ns}$, a scale much smaller than the $\mu$s time scale for the reverse-annealing protocol. We conclude that the parameter space of the DW-2000Q device, does not allow to access the regime that we discuss in this paper. Nevertheless, these limitations, in principle, might be mitigated by hardware alterations. The more recent quantum annealing device, the D-Wave Pegasus, may allow for simpler wirings than those that we propose for the implementation in the DW-2000Q. And perhaps access to features beyond those available to the cloud users may allow for probing shorter time scales that could permit the observation of the spinon localisation that we predict in our work.

## B  Analytic expression for the asymptotic long time spinon density and semi-analytical displacement of the spinon

In the long time limit, the wave function squared of a quasiparticle on the $\mathbb{Z}_2$ chain can be obtained analytically. The presence of randomly distributed static visons causes destructive interference in the propagation of the spinon. Therefore, the visons cut the chain into disconnected line segments for the spinon. Random distribution of visons means that the line segments for the spinon are sampled from the distribution

$$P(l,r) = \frac{1}{2^{l+1}} \frac{1}{2^{r+1}}, \tag{B.1}$$

where $l$ and $r$ are distances from the initial position to the left and to the right, respectively.

At long times, the thermal bath causes dephasing and thermalisation of the initial state, resulting in a uniform spinon density across each disconnected segment. Summing over segments of appropriately distributed lengths containing the initial position of the quasiparticle, we are then able to reconstruct the wave function squared

$$|\Psi(x)|^2 = \sum_l^\infty \sum_r^\infty \frac{1}{2^{l+r+2}} \underbrace{\frac{1}{l+r+1} \Theta(-l \le x \le r)}_{\left|\Psi^{(l,r)}(x,t)\right|^2},$$

where $\Theta(x) = 1$ if $x \in \langle -l, r \rangle$ and zero otherwise, and $\left|\Psi^{(l,r)}(x,t)\right|^2$ denotes the spinon density corresponding to the disconnected segment of length $l + r + 1$.

In Fig. 13 we show a comparison of the long time asymptotic spinon density, (absolute value square of its wave function) obtained from the numerics and from the analytical calculation over disconnected segments, Eq. (B.1), discussed above. There are no appreciable differences as indeed we expect the disconnected segment approximation to be asymptotically exact in the long time limit.

In the main text we also show a semi-analytical solution for the displacement of the spinon on the $\mathbb{Z}_2$ chain of length $L$ as a function of time, see dash-dotted black line in Fig. 4. In this case, we first calculate numerically the spinon density $\left|\Psi^{(l,r)}(x,t)\right|^2$ over a disconnected segment by evolving the initial wave function, and then average over the segment length distribution discussed above,

$$|\Psi(x,t)|^2 = \sum_{l}^{L/2} \sum_{r}^{L/2} \left|\Psi^{(l,r)}(x,t)\right|^2 P(l,r). \tag{B.2}$$

The vison displacement is then readily obtained as

$$\langle x(t)^2 \rangle = \sum_{l}^{L/2} \sum_{r}^{L/2} \sum_{j=-l}^{r} x_j^2 \left|\Psi^{(l,r)}(x_j,t)\right|^2 P(l,r). \tag{B.3}$$

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
