# Peer review of "Probing fractional statistics in quantum simulators of spin liquid Hamiltonians"

_SciPost Physics, doi:SciPost Phys. 15, 194 (2023)_

## Round 2 · Referee Report · Yaroslav Herasymenko (Referee 1) · 2023-1-12

Strengths

  1. The goal to probe anyonic statistics in a D-Wave type device is well-chosen both from the perspective of strongly correlated physics as well as for the development of quantum simulation techniques

  2. The authors suggest an original toy model that matches the hardware requirements remarkably well, and is adapted to provide a strong signature of the fractional statistics in spite of the difficulties of noisy simulations

  3. The effects of imperfections were thoroughly studied, and the connection to the real-world parameters are made

Weaknesses

  1. A specific protocol to initialize the experiment and extract the observables is not given or discussed

  2. Many parameter choices are given without sufficient discussion, although they are important to interpret the results and their consequences

  3. Lack of clarity in multiple places

Report

In my opinion, the Journal Criteria are met, in particular

(a) this work fits fairly well in the expectation categories 3 and 4 ("Open a new pathway..." and "Provide a novel and synergetic link...").

(b) the general acceptance criteria are basically met, with some caveats. p.1 "Be written in a clear and intelligible way..", p.3 "Provide sufficient details..", and p.5 "Provide all reproducibility-enabling resources" call for a bit of revision -- see "requested changes".

In general, the work left a pretty good impression on me in terms of its quality, creativity, and how well-chosen the goal is. Hence, I recommend it for publication upon a minor revision. (some writing is needed but no extra research)

Requested changes

Here are the changes I request, categorized by degree of priority:

Crucial

  1. A specific protocol to extract the features in <x^2> doesn’t seem to be proposed or discussed. The experiment must be somehow initialized and a specific observable read out in some particular way. It’s not explained which control capabilities of D-Wave or a more abstract set-up have to be used for this and how exactly. It might be easy to argue, but it’s crucial to have such an argument in writing.

  2. Decoherence 2a. How little coherence, or how much dephasing should be present for the experiment to succeed and demonstrate the claimed signature of fractional statistics? In my opinion, giving a specific number (e.g. in terms of model parameters) here is crucial for the completeness of “device coherence testing” claims. 2b. What type and magnitude of decoherence do the authors expect on D-Wave, e.g. from available experimental studies performed on D-Wave? This should be commented on since it determines the expectations from the experiment. 2c. Not clear why only dephasing is considered in the modeling. E.g. are there no expected “T1” effects or they are not important, and if so, why? This needs to be commented on in the paper, for completeness.

  3. It’s worth saying more explicitly what parameter ranges are available on D-Wave. On p. 11 it simply says “we propose J^DW=1 which corresponds to J=2.43 GHz” etc., and then the effective Gamma=0.14 GHz is compared to the D-Wave temperature of 0.27 GHz (with unfavourable outcome). It is not discussed what are the limitations to set J and K frequencies to different values and what consequences this has. There is only a vague reference to “the parameter space of the DW-2000Q device, at least in the version that is accessible on the cloud”. This is not nearly specific enough given the important role these statements have for the practical simulation.

Important

  1. Dephasing rate gamma=Gamma/2 is incredibly strong. Is this model consistent with the effective Hamiltonian dynamics of spinons which supposedly implies full coherence at least on the scale of a few sites? Be it a yes or a no, this point in my opinion warrants a sentence or two of comments. It seems that depending on this point, the disorder models and the numerical results have quantitative or only qualitative consequences for the experiment. The status of the consequences should also be stated more clearly.

  2. Effective mapping from CGS to Z2 ladder is very non-transparent. 2a. The proof/argument for it is not given/not referenced. It’s not even quite clear in which sense this effective mapping is claimed to hold (in which parameter regime of CGS model? what degrees of freedom are retained? what physics is lost in the mapping?). So, these should be clarified in the text. 2b. It’s also not clear what role this mapping plays in the rest of the analysis. It seems to me that its main purpose is to extract the spinon hopping amplitude (which is easier related to the parameter Gamma of the Z2 ladder than the parameters of CGS model). Is this correct? If that’s the case, it wasn’t announced or motivated very clearly in my opinion; and if it’s used for something more or something else, I failed to see it, hence other readers might fail to see it too. So, in the text, it would be useful to have a clear explanation of the role played by this mapping.

  3. If visons in the model correspond to G_p=-1 (page 3), and G_p are conserved, then how can visons have quantum dynamics (as suggested in many parts of the text)? This is perhaps simply my confusion, so it's sufficient to give a response to me directly and also to clarify this point in the text so that other readers aren't confused as well.

  4. More explicit motivation for the study of fractional statistics is needed. While I agree it is interesting, it is not explicitly explained what specific new physics knowledge or technical know-how could be gained from attempting a simulation of this specific toy model. In my opinion, it is important for the paper to have a few sentences explaining that. (Beyond the generic “paves the way” claims, of course – those aren't really useful unless one clearly states what intermediate steps are taken toward what specific goal.)

Less important

  1. It’s worth defining semion statistics and explaining why it is considered fractional. E.g. one can explicitly write that phase pi for a spinon looping around a vison means pi/2 for a simple exchange. Right now the text says “mutual statistical phase of π”, and some readers might be confused as to how this is different from fermions.

  2. “... the striking difference between the behaviour of spinons in presence or absence of visons can clearly be used as signature of …” – at the end of Sec. B. It is not clearly stated how “presence or absence of visons” is to be realized in the context of this test. I presume it’s the effective presence or absence by using the pinning field (and thus removing the interference), but I think it’s worth to say it explicitly.

---

## Round 2 · Referee Report · Anonymous (Referee 2) · 2023-2-2

Strengths

1 The idea of using quantum simulators, especially D-wave devices, for realizing and investigating quantum spin liquid is promising.
2 It is also promising that the proposed method can also be used to benchmark the devices to a certain extent.
3 Many numerical simulations to support the claims.

Weaknesses

1 The motivation is rather unclear; what would be the purpose of using a quantum simulator when we already know what it would provide?
2 The parameters used in numerical simulations are provided without much explanation.
3 It is unclear that the method proposed by this work can be actually realized experimentally using e.g. Dwave devices.

Report

I think the paper meets acceptance criteria (Expectations 3) and will meet general acceptance criteria after some minor revision.

Requested changes

1 The purpose of using a quantum simulator when we already know what it would provide was not clear to me. The approximations that the authors use to analyze the system seems reasonable enough, and I can expect that solving the full model Hamiltonian (by quantum simulators or maybe by a very powerful classical machine) would yield basically the same result. This point should be elaborated to certain extent.

2 It is assumed that the spinons only experience dephasing and not other types of error. Is it a realistic assumption?

3 Is $\Gamma$ in Eq. (3) and $\Gamma$ in Eq. (5) same quantity? If not, what would be the relationship of them?

4 In the numerical simulation, it seems to me that they are assuming (1) to be able to prepare the spinon vaccum state, and (2) the ability to excite a spinon at a single site, on an actual device. Is it possible?

5 They introduce disorder to the effective spinon Hamiltonian (Eq. (5) and Eq. (6)). But when experimentally realizing the proposed model, I think the disorder will be introduced to the coefficients of the base model (Eq. (1)). The connection between these are not clear.

6 About the comparison with the classical random walk (RW). I did not quite get why $\langle x^2\rangle$ of RW does not saturate at the same level as Z2 model in Fig. 5. This indicates in classical RW the hopping is allowed at any edges in contrast to the Z2 model where visons prohibits hopping of spinons at aroud half of the edges. Why do we not use the same connection as in the Z2 model but allow the hopping in RW?

7 It is claimed that the model should be useful for benchmarking "quantumness". To what extent this model would be useful in testing such a property compared to many other possible methods? For example, maybe we can just use a Hamiltonian dynamics $U=\exp(-iHt)$ and its inverse $U^\dagger=\exp(iHt)$ and measure something like $|\langle 0|U^\dagger U|0\rangle|^2$ which should be 1 when quantum coherence is preserved through the process but otherwise very small.

8 A minor comment: They use $\rho_{ss'}$ to express the state in Eq. (4) but in other places they use $|\Psi(x,t)\rangle$. This should be made consistent.

---

## Round 3 · Referee Report · Yaroslav Herasymenko · 2023-9-7

Strengths
As per previous report
Weaknesses
The authors improved on all key weaknesses mentioned in my previous report to the point I consider satisfactory
Report
Criteria are met
Requested changes
None, my main requests were satisfied
(although do note a typo -- "vision" instead of "vison" in Section V, can use ctrl-F to find it)
Congrats to the authors on the improved version of their paper. It's green light to publish on my end

---

## Round 3 · List of Changes

- We added in Appendix A several paragraphs to explain in detail the reverse annealing protocol. Specifically 1) how to initialise a state with a single spinon in D-Wave devices; 2) how to extract $\langle \, x^2 \, \rangle$ from the experimental data; 3) D-Wave's annealing schedule and its limitations on setting the coupling parameters.
- A new section explaining our claims of testing quantumness has been added before the conclusions.
- A brief explanation of dephasing and dissipation in the effective spinon propagation model, and the corresponding choice of value of $\gamma$, has been added at the start of Sec. III.
- We modified the hopping amplitude in Eq.~3 to be $2\Gamma$ to capture the effect that a spinon can hop through both top and bottom legs in a ladder, and changed Fig.~8 and Fig.~9 correspondingly.
- Added a footnote explaining the exchange statistics between a spinon and a vison in Sec. II.
- Added a paragraph summarising what each section is about in the introduction.
- We added Sec. IV B and discussed the limit in which we can identify the low-energy quasiparticle excitations, spinons and visons, in the two models, and their mutual semionic statistics.

You are currently on this page

---

## Editorial Decision

published